# Long-term trends in the incidence of peritoneal dialysis-related peritonitis disclose an increasing relevance of streptococcal infections: A longitudinal study

Joana Eugénio Santos[1], Catuxa Rodríguez Magariños[2], Leticia García Gago[2], Daniela Astudillo Jarrín[2], Sonia Pértega[3], Ana Rodríguez-Carmona[2], Teresa García Falcón[2], Miguel Pérez Fontán[2,4] *

1 Centro Hospitalario Espirito Santo, Evora, Portugal, 2 Division of Nephrology, University Hospital A Coruña, A Coruña, Spain, 3 Division of Epidemiology, University Hospital A Coruña, A Coruña, Spain, 4 Health Sciences Faculty, University of A Coruña, A Coruña, Spain

☯ These authors contributed equally to this work.
* miguel.perez.fontan@sergas.es

**Data Availability Statement:** Data cannot be shared publicly due to legal restrictions related to

## Abstract

### Background

The selective impact of strategies for prevention of PD-related peritonitis (PDrP) may have modified, in the long term, the causal spectrum, clinical presentation and outcomes of these infections.

### Objectives

To compare trends in the incidence of PDrP by different microorganisms during a 30-year period, with a particular focus on streptococcal infections. To analyze the clinical presentation and outcomes of these infections. Secondarily, to investigate how the isolation of different species of streptococci can influence the clinical course of PDrP by this genus of bacteria.

### Method

Following a retrospective, observational design we investigated 1061 PDrP (1990–2019). We used joinpoint regression analysis to explore trends in the incidence of PDrP by different microorganisms, and compared the risk profile (Cox), clinical presentation and outcomes (logistic regression) of these infections.

### Main results

Our data showed a progressive decline in the incidence of PDrP by staphylococci and Gram negative bacteria, while the absolute rates of streptococcal (average annual percent change +1.6%, 95% CI -0.1/+3.2) and polymicrobial (+1.8%, +0.1/+3.5) infections tended to increase, during the same period. Remarkably, streptococci were isolated in 58.6% of polymicrobial infections, and patients who suffered a streptococcal PDrP had a 35.8% chance of

data protection acts. However, they can be requested and obtained through the Galician Health System (Servicio Galego de Saúde, Sergas) vis email at ceic@sergas.es.

**Funding:** The authors received no specific funding for this work.

**Competing interests:** The authors have declared that no competing interests exist.

presenting at least one other infection by the same genus. The risk profile for streptococcal infections was comparable to that observed for PDrP overall. Streptococcal PDrP were associated with a severe initial inflammatory response, but their clinical course was generally nonaggressive thereafter. We did not observe a differential effect of different groups of streptococci on the clinical presentation or outcome of PDrP.

## Conclusions

Time trends in the incidence of PDrP by different microorganisms have granted streptococci an increasing relevance as causative agents of these infections, during the last three decades. This behaviour suggests that current measures of prevention of PDrP may not be sufficiently effective, in the case of this genus of microorganisms.

## Introduction

Peritonitis (PDrP) represents one of the most feared complications of chronic Peritoneal Dialysis (PD), associating significant rates of mortality [1,2] and PD technique failure [3–6]. The incidence of PDrP declined markedly between 1985 and 1995, after the introduction of Y-set, double bag systems, but improvements have been slower thereafter [5,7], and rates as high as one episode every two patient-years are still considered acceptable by current standards [8]. Singular advances, including the introduction of low glucose degradation products-based (low-GDP) solutions, may have been beneficial, but their impact has been disappointing, in clinical terms [9]. On the contrary, comprehensive approaches to prevention, including continuous quality improvement strategies, have provided a renewed hope of progress [10], and may mark the route to an optimized control of these infections.

Staphylococci are still viewed as the main source of PD-related PDrP [1,11,12]. However, a selective impact of preventive measures, preferentially oriented to reduce the consequences of touch contamination and catheter-related infections, may have influenced the etiologic spectrum and the strategies of management of PDrP [8], because these measures do not protect evenly from infections by different microorganisms. In particular, streptococcal infections have been a subject of limited attention in the past, due to the perceptions that these infections are relatively infrequent (with reported relative incidences of 5–12% of all PDrP) and follow a relatively benign clinical course [13–16]. On the other hand, the taxonomy and nomenclature of the ubiquitous genus *Streptococcus* has been reviewed in the last years [17] and, aside from single case reports, information on the compared aggressiveness of PDrP by different members of this family of microorganisms is scarce [18]. Finally, streptococci may be a common component of polymicrobial PDrP, but the real incidence of this circumstance and its clinical significance are also unclear.

Following an observational, retrospective design, we have investigated time trends in the incidence of PDrP in our centre during a period of 30 years. Our hypothesis was that advances in the prevention and management of PDrP could have modified the etiologic and clinical spectrum of PDrP, with potential consequences for the management and outcomes of these infections.

## Method

### General design

Following an observational, retrospective design, we analyzed trends in the incidence and outcomes of PDrP diagnosed in our centre between January 1990 and December 2019. Our main

focus was to disclose changes in the incidences of PDrP by different causative agents, with a particular interest in streptococcal infections (primary objective). For this purpose, we categorized the study follow-up into six periods (1990–94, 1995–99, 2000–04, 2005–09, 2010–14 and 2015–19). We also analyzed the clinical presentation, treatment and outcomes of the aforementioned groups, with streptococcal infections again as the main reference. The main clinical outcome was a composite of PDrP-related death, drop-out to hemodialysis for more than 3 months and need for catheter removal to control the infection (treatment failure). Secondarily, we investigated the clinical significance of PDrP by different species of streptococci, as also the role of these bacteria in polymicrobial infections (secondary objectives).

The study complied with the principles of the Declaration of Helsinki and the ethical requirements of our centre for retrospective observational studies. The study protocol was evaluated and approved by the local Ethical Committee of A Coruña-Ferrol Hospitals (code 2020/190). Oral informed consent was a requisite for inclusion of the minority of eligible individuals who were accessible at the time of initiation of the study, and conveniently registered in the clinical records of the patients. In the case of patients not available for consent (due to demise or loss to follow-up), inclusion was approved by the above mentioned Ethical Committee.

## Study population

We recruited for this study all patients starting PD in our unit between January 1, 1990 and October 1, 2019. Follow-up was closed by December 31, 2019. We excluded from analysis patients <18 years of age, those with a follow-up on PD <1 month, and those with inadequate clinical records. No patient denied consent for participation.

## Study variables and strategy of analysis

Our interest was addressed to episodes of PDrP occurring during the study period. In particular, we focused on streptococcal infections, and used PDrP of other causes as control groups. For this purpose, we categorized the following study groups:

1. *Streptococcus spp*.

2. *Staphylococcus aureus* (SAu).

3. Coagulase-negative staphylococci (CNSt).

4. *Enterobacteriaceae* and nonfermenting Gram negative bacteria (GNB).

5. Polymicrobial PDrP, excluding those with a rampant intestinal origin (immediate surgical approach or isolation of multiple enteric microorganismos, including intestinal anaerobics)

6. Culture-negative infections.

We did not consider PDrP with primary isolation of yeasts (n = 30), filamentous fungi (n = 2) or mycobacteria (n = 5), due to their low frequency and particular significance. PDrP caused by other less frequent (less than 50 cases) Gram positive [e.g. *Enterococcus spp* (n = 35) or *Corynebacterium spp* (n = 13)] or Gram negative bacteria [e.g. *Neisseria spp* (n = 10)] were also excluded from analysis. Only 4 episodes of PDrP by SAu (7.0%) were caused by methicillin-resistnat strains, and this small subset was not further individualized for analysis.

The main study variable was the causative organism of each PDrP, always with a main focus on streptococcal infections. We explored:

1. The time course of the incidence of PDrP by different causative microorganisms.

2. The risk profile for different types of PDrP, according to the demographic and clinical characteristics of the study population: age, gender, underlying kidney disease, diabetes mellitus, previous episodes of PDrP, treatment with immunosuppressants, PD vintage, PD modality, use of icodextrin, SAu carriage [19], comorbidity score (Charlson), malnutrition (subjective global assessment), socioeconomic status, assisted versus autonomous PD, body mass index (weight/height$^2$), hemoglobin, albumin (autoanalyzer), C-reactive protein (Immunoturbidimetry), glomerular filtration rate (GFR)(mean of urea and creatinine renal clearances), and peritoneal transport (D/P ratio of creatinine at 240´ D/Pcrea).

3. Clinical presentation of the PDrP: simultaneous catheter-related infection (catheter-related PDrP), hospitalization, peritoneal cell-count at baseline, and number of days until complete clinical, bacteriologic and cytologic remission or catheter removal.

4. Initial and susceptibility-oriented antimicrobial therapy

5. Outcomes of PDrP:

- Peritoneal catheter removed

- Transfer to hemodialysis for at least 3 months after the infection

- Death for any reason during hospital admission or within 30 days after initiation of the episode.

- Treatment failure, defined by any of the three previous outcomes (main clinical outcome).

PDrP was defined according to the ISPD guidelines [8,20]. The same sources were used for definition of relapsing, recurrent and repeat PDrP. The initial diagnostic procedures (including systematic sampling of dialysate for cytologic and microbiologic evaluation) have remained essentially stable during the whole study period. Our protocol for initial treatment of PDrP was based on intraperitoneal ciprofloxacin between 1990 and 2007. In September 2007, we switched to intravenous vancomycin + intraperitoneal cefotaxime, after susceptibility of CNSt to ciprofloxacin fell below acceptable standards. In our Unit, hospital admission for PDrP is indicated for high-risk patients, infections with an aggressive clinical presentation, treatment failure after oriented antimicrobial therapy and isolation of microorganisms demanding in-centre management. Outpatient management includes a clinical check-up every other day until full remission of infection, and then antibiotic therapy during 2–3 weeks, according to the ISPD recommendations [8,20].

We used integrated Y systems for continuous ambulatory PD, and Home Choice (Baxter, Deerfield, IL, USA) cyclers for automated PD. Conventional, lactate-based PD solutions were used until March 2008, when patients were switched to low-GDP solutions. We have performed systematic screening of nasal (since 1990) and pericatheter (since 1996) carriage of SAu to patients and PD partners during the whole study period. Carriers were treated with nasal and pericatheter mupirocin.

**Statistics.** Continuous variables are expressed as mean ± SD, or as median (interquartile range), in case of markedly non-normal distribution. Categorical variables are presented as the number of cases (%). Basic univariate comparisons were produced by means of usual parametric [two-tailed Student's t test, ANOVA (Scheffé)] and nonparametric tests ($\chi^2$ distribution, Mann Whitney, Spearman's correlation coefficient), as needed.

We used joinpoint regression analysis to disclose time trends in the overall and by-causative agent rates of incidence of PDrP, calculating the annual percent change (APC), and the average annual percent change (AAPC) for the whole study period [21]. The risk profile for PDrP by different families of microorganisms (survival to the first episode) was first explored by

means of Kaplan Meier plots (log rank test), and then by stepwise Cox models. For Kaplan Meier analyses, we categorized numerical variables by tertiles. For multivariate analyses, we considered only first order interaction terms. PD vintage (categorized in three periods: 1990–99, 2000–09 and 2010–19) was systematically considered as a control variable in these multivariate models.

The clinical presentation and antimicrobial therapy characteristics are presented using a descriptive approach, including univariate comparisons. The same applies for outcomes, in which case we also carried out a multivariate approach (stepwise logistic regression), to categorize better the role of different bacteria, on these outcomes. Risk estimations are presented as Odds ratios (OR) (logistic regression) or hazard ratios (HRs)(Cox), with 95% confidence intervals (CI). A two-tailed p-value <0.05 was considered significant (the software used for time trends analyses does not provide exact p values).

We used the IBM-SPSS 19.0 software for general data management and statistic analyses. Time trends analyses were produced with the help of the Joinpoint Regression Program 4.7.0.0. (Feb 2019; Statistical Research and Applications Branch, National Cancer Institute).

## Results

### Overview and time trends

The study population included 878 patients, after excluding 25 other individuals who did not meet the inclusion criteria. Their main baseline characteristics are presented in Table 1. Of the whole study group, 468 patients (52.6%) suffered at least one episode of PDrP, and 158 (18.0%) suffered at least one streptococcal PDrP. The latter was the etiologic group with the highest incidence of repeat infection (35.8%). Thirty (19.0%) of these patients suffered two, and 25 others (15.8%) more than two monobacterial streptococcal PDrP during follow-up (range 3–7 episodes). Only CNSt approached this figure (29.1%, including 12.6% with more than two episodes), which was markedly lower in the cases of SAu (11.6%), GNB (21.2%), polymicrobial (11.1%) and culture-negative PDrP (16.0%)(p = 0.003). Median delays to the first episode of PDrP were 8 (interquartile range 3 to 17) (overall) and 12 months (interquartile range 5 to 24)(streptococcal PDrP).

Overall, we recorded 1061 episodes of PDrP during follow-up, caused by streptococci (n = 235, 22.1%), SAu (n = 57, 5.4%), CNSt (n = 285, 26.9%) or GNB (n = 175, 16.5%). We also registered 133 polymicrobial (12.5%) and 126 (11.9%) culture-negative infections. Regarding streptococcal infections, 223 were caused by viridans group streptococci and only 12 by non-viridans species. Among viridans streptococci, the *mitis* (including *oralis*, *sanguis and gordonii*)(n = 92) and the *salivarius* (including *vestibularis*)(n = 85) groups predominated. On the other hand, at least one streptococcal strain was isolated in 78 polymicrobial infections (58.6%) (Table 2). Streptococci were less frequently identified in the latter group between 1990 and 1999 (38.9%), but kept consistently above 60% after 2000 (p = 0.016).

Table 3 displays the absolute and relative incidences of PDrP during the different study periods. Joinpoint analyses disclosed a progressive decrease in the incidence of PDrP overall. This tendence was supported by a declining incidence of infections by staphylococci and GNB (Table 3, Fig 1), while an opposite trend was observed for polymicrobial and streptococcal infections (not significant in the latter case) (Fig 1). Separate APC analyses showed rather homogeneous time courses for these trends, except in the cases of PDrP overall [AAPC -5,4% per year 1990–1999 (95% CI -9.0/-1.6, p<0,001) versus -0.7% per year 2000–2019 (95% CI -1.7/+0.2)] and infections by *Staphylococcus aureus* [AAPC -12.4% per year 1990–1999 (95% CI -22.0/-1.6), 17.5% per year2000-2008 (95% CI -1.9/+40.8) and -21.7% per year 2009–2019 (95% CI -31.3/-10.7)].

**Table 1. Baseline characteristics of the study population.**

| | |
|---|---|
| Age (years) | 59.2 (15.6) |
| Gender (males/females)(%) | 522/356 (59.5/40.5) |
| Kidney disease | |
| Glomerular | 117 (13.3) |
| Interstitial | 97 (11.0) |
| Vascular (including nephroangiosclerosis) | 84 (9.6) |
| Cystic | 76 (8.7) |
| Systemic diseases (including paraproteinemias) | 46 (5.2) |
| Diabetic nephropathy | 264 (30.0) |
| Other/Unknown | 194 (22.1) |
| Diabetes (%) | 310 (39.4) |
| Charlson's score | 4,1 (2.0) |
| Recent/Ongoing immunosuppression (%) | 85 (9.7) |
| Malnutrition (%) | 83 (9.4) |
| Body mass index (Kg/m$^2$) | 26,0 (4.9) |
| Low socioeconomic status (%) | 281 (32.0) |
| Family-assisted PD (%) | 347 (39.6) |
| PD vintage (start 1990–94; 1995–99; 2000–04; 2005–09; 2010–14; 205–19)(%) | 134/192/178/135/123/116 (15.3/21.9/20.3/15.4/ 14.0/13.2) |
| Modality of PD (Automated PD/CAPD)(%) | 274/604 (31.2/68.8) |
| Icodextrin (%) | 184 (21.0) |
| D/P creatinine 240', baseline PET (n = 664) | 67,9 (12.1) |
| *Staphylococcus aureus* carrier (%) | 383 (43.6) |
| Glomerular filtration rate (mL/minute) | 6,2 (4.1) |
| Hemoglobin (g/dL) | 10,5 (1.6) |
| Plasma albumin (g/L) | 37,1 (5.6) |
| Serum C reactive protein (mg/dL)(n = 717) | 0.59 (0.28–1.45) |
| Follow-up (months) | 28.9 (25.7) |

Figures denote mean values ± standard deviation, median with interquartile range (numerical variables) or absolute numbers (%) (categorical variables)

Keys: PD: Peritoneal Dialysis; CAPD: Continuous Ambulatory Peritoneal Dialysis; PET: Peritoneal equilibration test

## Risk profile for PDrP

Kaplan Meier analysis disclosed correlations between the probability of presenting PDrP (any cause) during follow-up, on one side, and older age (p = 0.023 log rank), higher Charlson's score (p = 0.008), lower GFR (p = 0.035), higher C-reactive protein levels (p = 0.012), higher body mass index (p = 0.035), and family-assisted PD (p = 0.045), on the other. Patients on automated PD also exhibited a trend to a lower risk of infection than those on CAPD

**Table 2. Microorganisms isolated in polymicrobial peritonitis.**

| Microorganism | n |
|---|---|
| Coagulase-negative staphylococci | 55 |
| *Staphylococcus aureus* | 7 |
| *Streptococcus spp* | 116 |
| *Enterococcus spp* | 23 |
| *Enterobacteriaceae* and nonfermenting Gram negative bacteria | 83 |
| Other | 8 |

**Table 3. Time course of the absolute and relative rates of incidence of PD-related peritonitis of different etiologies.**

|  | 90–94 | 95–99 | 00–04 | 05–09 | 10–14 | 15–19 |
|---|---|---|---|---|---|---|
| Follow-up (months) | 3558 | 4748 | 5495 | 4823 | 4182 | 4442 |
| Follow-up (patient-years) | 296.5 | 395.7 | 457.9 | 401.9 | 348.5 | 370.2 |
| **ABSOLUTE RATES** (episodes/patient/year) |  |  |  |  |  |  |
| **All infections** (n) | 190 | 188 | 200 | 183 | 154 | 146 |
| Incidence | 0.64 | 0.48 | 0.44 | 0.45 | 0.44 | 0.39 |
| *Streptococcus spp* (n) | 31 | 25 | 50 | 36 | 48 | 45 |
| Incidence | 0.104 | 0.063 | 0.109 | 0.089 | 0.138 | 0.122 |
| *Staphylococcus aureus* (n) | 13 | 10 | 7 | 21 | 4 | 2 |
| Incidence | 0.043 | 0.025 | 0.015 | 0.052 | 0.001 | 0.005 |
| **Coagulase negative** *Staphylococcus spp* (n) | 69 | 59 | 59 | 51 | 26 | 21 |
| Incidence | 0.233 | 0.149 | 0.129 | 0.127 | 0.074 | 0.057 |
| **Gram negative bacteria*** (n) | 29 | 35 | 32 | 32 | 26 | 21 |
| Incidence | 0.098 | 0.088 | 0.070 | 0.079 | 0.074 | 0.057 |
| **Polymicrobial** (n) | 11 | 25 | 24 | 17 | 26 | 30 |
| Incidence | 0.037 | 0.063 | 0.052 | 0.042 | 0.074 | 0.081 |
| **Negative culture** (n) | 14 | 23 | 24 | 19 | 22 | 24 |
| Incidence | 0.047 | 0.058 | 0.052 | 0.047 | 0.063 | 0.065 |
| **RELATIVE RATES** (% of all episodes of PDrP) |  |  |  |  |  |  |
| *Streptococcus spp* | 16.3 | 13.3 | 25.2 | 19.7 | 31.2 | 31.0 |
| *Staphylococcus aureus* | 6.8 | 5.3 | 3.5 | 11.5 | 2.6 | 1.4 |
| **Coagulase negative** *Staphylococcus spp* | 36.3 | 31.6 | 29.7 | 28.0 | 16.9 | 14.5 |
| **Gram negative bacteria*** | 15.3 | 18.7 | 16.0 | 17.5 | 16.9 | 14.5 |
| **Polymicrobial** | 5.8 | 13.3 | 12.0 | 9.3 | 16.9 | 20.7 |
| **Culture-negative** | 7.4 | 12.3 | 12.0 | 10.4 | 14.3 | 16.6 |

* *Enterobacteriaceae* + Nonfermenting Gram negative bacteria

(p = 0.07). On the other hand, the risk of suffering streptococcal PDrP was univariately linked to older age (p = 0.003), lower plasma albumin (p = 0.041), higher body mass index (p = 0.021), lower blood haemoglobin (p = 0.047) and automated PD rather than CAPD (p = 0.050). PD vintage was a univariate predictor of the risk of streptococcal (p = 0.004), but not overall PDrP (p = 0.124)(other variables displayed in Table 1 not significant).

Table 4 shows the multivariate (Cox) risk profile for PDrP overall, and streptococcal PDrP. Both models were quite similar, except for a clearly stronger impact of PD vintage on the risk of streptococcal infection, and a more marked association of baseline GFR with the later risk of PDrP overall.

## Clinical presentation according to causative microorganism

The main clinical characteristics of PDrP by different causative microorganisms are displayed in Table 5. We did not record a single instance of streptococcal catheter-related PDrP between 1990 and 2019.

Streptococcal PDrP associated the most severe initial inflammatory reaction of all the study groups, as estimated from the baseline peritoneal cell count and the proportion of polymorphonuclear leukocytes. Otherwise, the clinical presentation of streptococcal PDrP showed similarities to infections by CNSt and culture-negative PDrP. Final antimicrobial therapy of these infections was based on intraperitoneal ciprofloxacin (until 2007), cephalosporins and vancomycin; only 15.0% of cases were finally treated with an antibiotic association.

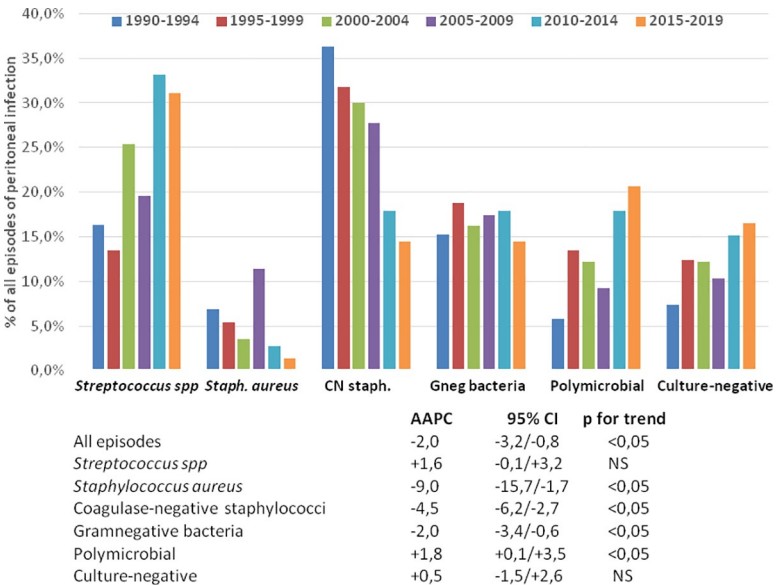

**Fig 1. Relative incidence of peritoneal infection by different causative microorganisms during six consecutive 5-year periods (upper), and Average Annual Percent Changes (AAPC) in the absolute rates of incidence during the 30-year study period.**

## Clinical outcomes

The main clinical outcomes of PDrP are presented in Table 6. Remarkably, streptococcal PDrP carried the highest probability of uneventful continuation of PD. Catheter removal was less frequent than in any other type of PDrP, except culture-negative PDrP (p = 0.04 versus CNSt). The risk of relapse, in opposition to the case of repeat infection (see above) was low, and treatment failure was less frequent than in any other group, although the difference with CNSt (p = 0.072) and negative-culture PDrP (p = 0.11) did not reach statistical significance.

Table 7 presents the results of multivariate analysis for the main outcomes. Streptococcal PDrP carried a better prognosis than infections by SAu or GNB, and similar to CNSt and culture-negative infections.

**Table 4. Risk profile for peritoneal infection (overall) and streptococcal peritonitis.** Multivariate analysis.

|  | Overall | | | Streptococcal | | |
|---|---|---|---|---|---|---|
|  | **HR** | **95% CI** | **P value** | **HR** | **95% CI** | **P value** |
| PD vintage (ref. 1990–99) |  |  |  |  |  |  |
| 2000–09 | 1.21 | 0.94–1.57 | 0.11 | 1.71 | 1.22–2.62 | 0.009 |
| 2010–19 | 0.97 | 0.72–1.32 | 0.94 | 1.77 | 1.12–2.80 | 0.015 |
| Age (per year) | 1.01 | 1.00–1.03 | 0.026 | 1.02 | 1.01–1.04 | 0.001 |
| Plasma albumin (per g/L) | 0.98 | 0.96–0.99 | 0.021 | 0.96 | 0.93–0.99 | 0.011 |
| GFR (per mL/min) | 0.97 | 0.94–0.99 | 0.029 | - | - | - |
| Body mass index (per Kg/m$^2$) | 1.03 | 1.00–1.05 | 0.021 | 1.04 | 1.00–1.07 | 0.046 |

Stepwise Cox's regression. First-order interaction terms not significant. For the overall risk of PDrP, -2log likelihood 4429.30, χ2 28.93, p<0.0005. For streptococcal PDrP, -2log likelihood 1692.86, χ2 36.52, p<0.0005

Keys: HR: Hazard Ratio; CI: Confidence interval; PD: Peritoneal Dialysis; GFR: Glomerular Filtration Rate

**Table 5. Clinical presentation and antimicrobial therapy.**

| | Strept. | SAu | CNS | GNB | Poly | Culture-negative | P value |
|---|---|---|---|---|---|---|---|
| Catheter-related (%) | 0 | **34.1** | **2.3** | **14.2** | 0.8 | 0 | 0.0005 |
| Hospitalization (%) | 14.0 | **40.4** | 13.3 | **33.0** | **26.7** | 143 | 0.0005 |
| Number of in-hospital days for admitted patients | 10.7 (9.4) | 16.3 (16.2) | 13.3 (10.2) | **20.5 (165)** | 12.9 (10.4) | 13.8 (13.1) | 0.045 |
| Initial peritoneal cell count (per mm$^3$) | 5431 (6995) | **2483 (2379)** | 2650 (5276) | 3466 (5463) | 3240 (3004) | **2151 (3431)** | 0.0005 |
| Initial neutrophil count (%) | 78.9 (12.3) | 74.5 (12.7) | **72.1 (15.4)** | 77.3 (14.0) | 79.6 (14.9) | **68.8 (18.3)** | 0.0005 |
| Initial treatment (%) | | | | | | | 0.0005 |
| Ciprofloxacin | 55.6 | 59.3 | **75.5** | **69.5** | **44.9** | 59.1 | |
| Vanco + Cephalosporin | 42.7 | 35.2 | **21.1** | **26.5** | **53.5** | 37.6 | |
| Other/No data | 1.7 | 5.5 | **3.4** | **4.0** | **1.6** | 3.3 | |
| Final treatment (%) | | | | | | | 0.0005 |
| Ciprofloxacin | 19.8 | **16.6** | **36.1** | **33.7** | **3.3** | 40.3 | |
| Cephalosporin | 28.3 | **5.6** | **4.7** | **23.3** | 164 | **6.0** | |
| Carbapenem | 1.4 | **1.9** | **0** | **8.4** | **4.9** | **2.7** | |
| Vancomycin | 33.8 | **18.5** | **36.0** | **0** | **6.6** | **4.7** | |
| Antibiotic association* | 15.0 | **55.6** | **22.4** | **28.9** | **66.9** | **44.3** | |
| Other/No data | 1.7 | **1.8** | **0.8** | **5.7** | **1.9** | **2.0** | |
| Number of days of antibiotic therapy | 15.6 (4.8) | **22.1 (8.0)** | 15.8 (5.9) | 16.6 (6.1) | **19.7 (5.2)** | **12.9 (5.1)** | 0.0005 |

* Most common antibiotic associations Vancomycin + Cephalosporin/Rifampicin (Gram positive microorganisms) or Cephalosporin + Aminoglycoside (Gram negative microorganisms)

Figures denote % of cases (categorical variables) or mean (standard deviation)(numerical variables). Comparison by χ2 distribution and one-way ANOVA. P values denote overall significance. In bold, categories presenting a significant difference with streptococcal infections (Scheffé)

Keys: Strept: Streptococci; SAU: *Staphylococcus aureus*; CNS: Coagulase-negative staphylococci; GNB: Gram negative bacteria (*Enterobacteriaceae* + Nonfermenting Gram negative bacteria); Poly: Polymicrobial

**Table 6. Clinical outcomes.**

| | Strept. | SAu | CNS | GNB | Poly | Culture-negative | P value |
|---|---|---|---|---|---|---|---|
| Time to remission or catheter removal (days) | 5.7 (3.9) | 5.3 (3.6) | 5.2 (3.5) | 5.1 (4.7) | 4.3 (2.2) | 5.0 (2.9) | 0.033 |
| Catheter removed (%) | 3.0 | **31.6** | **7.0** | **23.1** | **8.1** | 2.3 | 0.0005 |
| Relapse (%) | 10.7 | 8.8 | **17.9** | **19.8** | 9.5 | 7.6 | 0.003 |
| Recurrence (%) | | | | | | | 0.16 |
| Other bacteria | 0.4 | 0 | 2.1 | 4.4 | 0.7 | 1.8 | |
| Yeasts | 2.6 | 3.5 | 2.5 | 2.2 | 2.2 | 0.8 | |
| Main outcomes (%) | | | | | | | 0.0005 |
| PD continued >3 months | 97.0 | **76.9** | 96.1 | **85.2** | 91.9 | 94.0 | |
| Drop-out to Hemodialysis | 0 | **8.8** | 2.8 | **6.6** | 2.2 | 0.8 | |
| PDrP-related death | 3.0 | **12.3** | 1.1 | **8.2** | 5.9 | 5.3 | |
| Treatment failure (%) | 4.7 | **42.1** | 8.4 | **26.9** | **11.1** | 7.5 | 0.0005 |

Figures denote % of cases (categorical variables) or mean (standard deviation)(numerical variables). Comparison by χ2 distribution and one-way ANOVA. P values denote overall significance. In bold, categories presenting a significant difference with streptococcal infections (Scheffé)

Keys: Strept: Streptococci; SAU: *Staphylococcus aureus*; CNS: Coagulase-negative staphylococci; GNB: Gram negative bacteria (*Enterobacteriaceae* + Nonfermenting Gram negative bacteria); Poly: Polymicrobial

**Table 7. Predictors of clinical outcomes.** Multivariate analysis.

| | OR | 95% CI | P value |
|---|---|---|---|
| **CATHETER REMOVAL** | | | |
| Age (per year) | 0.97 | 0.96–0.99 | 0.002 |
| Time on PD at the time of infection (per month) | 1.02 | 1.01–1.03 | 0.0005 |
| PD vintage (Ref. 1990–99) | | | 0.86 |
| 2000–2009 | 1.06 | 0.84–1.76 | |
| 2010–2019 | 0.78 | 0.36–1.43 | |
| Causative agent of infection (Ref. *Streptococcus spp*) | | | |
| *Staphylococcus aureus* | 10.85 | 4.09–18.82 | 0.0005 |
| Coagulase-negative *Staphylococcus spp* | 1.73 | 0.70–4.28 | 0.23 |
| Gram negative bacteria | 8.39 | 3.62–19.48 | 0.0005 |
| Polymicrobial | 2.19 | 0.81–5.92 | 0.12 |
| Culture-negative | 0.72 | 0.18–2.85 | 0.64 |
| **PERITONEAL INFECTION-RELATED DEATH** | | | |
| Age (per year) | 1.10 | 1.06–1.14 | 0.0005 |
| Time on PD at the time of infection (per month) | 1.03 | 1.02–1.04 | 0.0005 |
| Charlson's score (per point) | 1.15 | 1.03–1.33 | 0.019 |
| Recent/Active immunosuppression | 4.99 | 1.91–13.02 | 0.001 |
| PD vintage (Ref. 1990–99) | | | 0.80 |
| 2000–2009 | 1.53 | 0.70–3.36 | |
| 2010–2019 | 0.64 | 0.26–1.56 | |
| Causative agent of infection (Ref. *Streptococcus spp*) | | | |
| *Staphylococcus aureus* | 10.50 | 3.05–36.12 | 0.0005 |
| Coagulase-negative *Staphylococcus spp* | 0.25 | 0.06–1.02 | 0.054 |
| Gram negative bacteria | 2.74 | 1.03–7.30 | 0.041 |
| Polymicrobial | 1.66 | 0.55–5.03 | 0.36 |
| Culture-negative | 2.02 | 0.65–6.29 | 0.22 |
| **TREATMENT FAILURE** | | | |
| Time on PD at the time of infection (per month) | 1.02 | 1.01–1.03 | 0.0005 |
| PD vintage (Ref. 1990–99) | | | 0.45 |
| 2000–2009 | 1.22 | 0.78–1.92 | |
| 2010–2019 | 0.89 | 0.52–1.52 | |
| Causative agent of infection (Ref. *Streptococcus spp*) | | | |
| *Staphylococcus aureus* | 14,30 | 6.28–32.59 | 0.0005 |
| Coagulase-negative *Staphylococcus spp* | 1.41 | 0.66–3.00 | 0.38 |
| Gram negative bacteria | 6.90 | 3.41–13.90 | 0.0005 |
| Polymicrobial | 2.00 | 0.88–4.58 | 0.09 |
| Culture-negative | 1.76 | 0.98–2.31 | 0.22 |

Best models. Stepwise logistic regression analysis

Keys: OR: Odds ratio; CI: Confidence interval; PD: Peritoneal Dialysis; Treatment failure: At least one of: catheter removal, death or drop-out to hemodialysis for at least 3 months

## Comparisons of PDrP by different species of streptococci

We compared the clinical presentation and outcomes of four different subgroups of streptococci, namely *mitis/oralis/sanguis/gordoni* (n = 92), *salivarius/vestibularis* (n = 85), other viridans (n = 46) and non-viridans (n = 12). We were unable to disclose any significant differences or trends, regarding the variables displayed in Tables 5 and 6 (data not presented).

### Clinical significance of the isolation of streptococci in polymicrobial PDrP

As previously stated, at least one streptococcal strain was isolated in 78 of the 133 polymicrobial PDrP included in the analysis. In 28 cases, polymicrobial PDrP was caused by 2 different strains of streptococci. Polymicrobial PDrP with presence of streptococci presented with marginally higher peritoneal cell counts (3501 vs 2764 cells/mm$^3$, p = 0.092) and % of neutrophils (81.9 vs 75.9%, p = 0.031), than infections without participation of these bacteria. We did not detect any other clinical difference, between these subsets (data not presented).

## Discussion

Our results provide clues to understand how the causative spectrum of PDrP may have evolved, during the last three decades. The systematic implementation of measures of prevention, essentially oriented to reduce touch contamination and catheter-related PDrP [8,22] resulted in a progressive decline of the incidence of infections by staphylococci and GNB (Table 3, Fig 1). On the contrary, in a setting of slowly decreasing incidence of PDrP, we observed trends to an increase in the rates of streptococcal and polymicrobial infections, with the particularity that a majority of the latter included isolation of streptococci. Overall, streptococci were isolated in 41.3% of the PDrP diagnosed between 2010 and 2019. Thinking in positive terms, this change may have been beneficial, because streptococcal PDrP associated better outcomes than infections by SAu or GNB (and similar to CNst) (Tables 6 and 7). From an opposite point of view, this evolution suggests that current prevention measures may not be effective, to reduce the risk of streptococcal infections. This finding should not be completely unexpected. First, catheter-related streptococcal PDrP is an infrequent event (Table 5) [23,24], downplaying the capacity of catheter care to prevent this complication. Touch contamination is a potential source of streptococcal infection, but the ubiquity (skin, mouth, upper respiratory tract, upper and lower gastrointestinal tract) of this genus of bacteria [17] entails more potential foci of contamination and, probably, a higher capacity for hematogenous spread than CNSt. The circumstance that 35.8% of the patients who presented streptococcal PDrP suffered at least one other episode of infection by the same genus suggests a persistence and/or multiplicity of foci of infection. Other studies have detected a significant incidence of repeat streptococcal PDrP, although not to the extent observed in our study [14,15]. The oral cavity is a subject of particular concern as a source of streptococcal PDrP, due to a high degree of colonization by this family of bacteria [25] and to a potential for both hematogenous spread and direct contamination, particularly if face masks are used inappropriately during the PD exchange [8]. On the other hand, the high proportion of polymicrobial PDrP with isolation of streptococci [13,14,16] suggests that the lower gastrointestinal tract may be another significant source of infections by these bacteria. Unfortunately, we were unable to create a specific risk profile for streptococcal PDrP (Table 4), partly due to the nonavailability of potentially relevant variables, including bucodental care, adherence to the use of face masks during the PD changes or intestinal disorders. Older age, hypoalbuminemia and overweight was associated with a higher risk of streptococcal PDrP, but this predictive model lacked specificity, because these factors were also linked to the general risk of PDrP [26–28]. This inability to establish a specific predictive model for streptococcal PDrP was also observed in the best powered study so far [14].

Our results disclosed a higher relative incidence of streptococci in both monobacterial and polymicrobial PDrP than previous reports [11–14,16,29–32]. However, most of the cited studies investigated patients started on PD in the 1990s', when the incidence of streptococcal PDrP was also lower in our centre. In addition, local factors, including differences concerning PD practices and the global incidence of PDrP, could help to explain this discrepancy. For instance, irregular adherence to measures with a demonstrated efficacy to prevent staphylococcal PDrP

[33] may associate a lower incidence of streptococcal infections [14,34,35]. The latter factor could also explain why only some studies [12,13,36] were able to detect the trend to an increase in the incidence of streptococcal PDrP, so clearly disclosed by our analysis. The exclusion by protocol of rampant enteric PDrP and the high proportion of polymicrobial peritonitis with isolation of streptococci may help to explain why, in our study, the observed outcomes of these infections were more benign than usually reported (Table 6) [29,31].

The present study agrees, for the most part, with previous reports on the clinical presentation and outcome of streptococcal PDrP (Tables 5 and 6). These infections have a relatively severe initial presentation, with a marked inflammatory response, as shown by the baseline peritoneal cell counts (Table 5), and a relatively slow resolution (Table 6) [13]. However, initial aggressiveness is usually followed by complete recovery with antibiotic therapy alone, with low rates of hospital admission, catheter removal, relapse, recurrence and, most importantly, hard outcomes (mortality, technique failure) (Table 7) [13–16,18]. Previous studies have shown that the time course of peritoneal cellularity, rather than the initial count, is the best marker of outcome of PDrP [37,38]. Our data also confirm that these bacteria are usually susceptible to common antibiotic regimes [39], as either vancomycin, betalactams or fluoroquinolones were able to control streptococcal PDrP (Table 5) [14,15]. Current ISPD recommendations endorse the use of ampicillin, for this purpose [8] but, in our unit, the success of the antimicrobials used for initial management of PDrP explains why ampicillin was not a usual choice.

Our results do not suggest particularities for any subgroup of streptococci, regarding clinical presentation or outcomes. Isolation of *Streptococcus bovis* has been linked to an increased risk of colorectal cancer. We recorded 5 instances of this infection, and all followed an uneventful clinical course, in agreement with a recent report [18]. We neither observed any significant diference among polymicrobial PDrP with or without isolation of streptococci. Previous studies have underlined that the isolation of GNB [31], intestinal anaerobics and *Enterococcus faecium* [40] may represent better markers of a complicated outcome of polymicrobial PDrP.

Our study suffers significant limitations, including a retrospective, single-centre design, which may overrate the effect of local factors. Some variables with a potential impact on the risk of streptococcal PDrP (e.g. oropharyngeal disorders) could not be recorded. Therapy used for treatment of streptococcal infections may have been broader than necessary. On the other side, significant strengths include the large time span, very appropriate to investigate time trends, and the high quality of our database, which permitted a complete analysis of the study population.

In summary, the last decades have contemplated an increasing relevance of streptococci in the causal spectrum of PDrP. This circumstance has been favoured by the concurrence of non-significant trends to an increase in the incidence of infections by this family of microorganisms, on one side, and simultaneous declines in the incidences of PDrP by staphylococci and GNB, on the other. This time course suggests that current measures for prevention of PDrP may not be sufficiently effective, for this genus of microorganisms. The facts that streptococcal PDrP associate a high risk of repeat infections and that streptococci are frequently isolated in polymicrobial infections reinforce this perception. Streptococcal PDrP are characterized by an initially aggressive clinical course and a relatively slow resolution, but also by a usually uneventful outcome with antibiotics alone. Future research will be necessary to improve prevention of infections by this genus of Gram positive bacteria.

## Author Contributions

**Conceptualization:** Joana Eugénio Santos, Catuxa Rodríguez Magariños, Leticia García Gago, Daniela Astudillo Jarrín, Sonia Pértega, Ana Rodríguez-Carmona, Teresa García Falcón, Miguel Pérez Fontán.

**Data curation:** Catuxa Rodríguez Magariños, Leticia García Gago, Daniela Astudillo Jarrín, Miguel Pérez Fontán.

**Formal analysis:** Leticia García Gago, Teresa García Falcón, Miguel Pérez Fontán.

**Investigation:** Joana Eugénio Santos, Catuxa Rodríguez Magariños, Sonia Pértega, Miguel Pérez Fontán.

**Methodology:** Sonia Pértega, Miguel Pérez Fontán.

**Software:** Sonia Pértega.

**Supervision:** Miguel Pérez Fontán.

**Validation:** Joana Eugénio Santos, Miguel Pérez Fontán.

**Writing – original draft:** Joana Eugénio Santos, Catuxa Rodríguez Magariños, Miguel Pérez Fontán.

**Writing – review & editing:** Sonia Pértega, Ana Rodríguez-Carmona, Teresa García Falcón, Miguel Pérez Fontán.

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
