## [Decision Letter · Decision Letter 0]

3 Nov 2020

PONE-D-20-22429

The Growing Role of Streptococci as Causative Agents of Peritoneal Dialysis-related Peritonitis. A Longitudinal Analysis

PLOS ONE

Dear Dr. Fontan,

Thank you for submitting your manuscript to PLOS ONE. After careful consideration, we feel that it has merit but does not fully meet PLOS ONE’s publication criteria as it currently stands. Therefore, we invite you to submit a revised version of the manuscript that addresses the points raised during the review process.  Those points are included below.

We look forward to receiving your revised manuscript.

Kind regards,

Sean Reid

Academic Editor

PLOS ONE

Journal Requirements:

2. Please amend your current ethics statement to address the following concerns: Please explain why written consent was not obtained, how you recorded/documented participant consent, and if the ethics committees/IRBs approved this consent procedure.

Reviewers' comments:

Reviewer's Responses to Questions

**Comments to the Author**

1. Is the manuscript technically sound, and do the data support the conclusions?

Reviewer #1: Yes

Reviewer #2: Partly

2. Has the statistical analysis been performed appropriately and rigorously? 

Reviewer #1: Yes

Reviewer #2: No

3. Have the authors made all data underlying the findings in their manuscript fully available?

Reviewer #1: No

Reviewer #2: Yes

4. Is the manuscript presented in an intelligible fashion and written in standard English?

Reviewer #1: Yes

Reviewer #2: Yes

5. Review Comments to the Author

Reviewer #1: Overview

The authors have conducted a 30-year retrospective chart review to evaluate trends in incidence of PD-related peritonitis. They have found decreasing trends in staphylococci and Gram negative bacterial infections and an increase in streptococcal and polymicrobial infection. There wasn’t a difference in predictive risk factors for Streptococcal PDrP but had more favorable outcomes. There were no differences amongst species of Streptococci. This is information that is useful to clinicians, infection prevention and improving quality of care. The authors have described the various aspects of this study succinctly and performed an excellent review of why this is an issue of practical importance. They also highlight, the potential difficulties with interpreting this data due to the retrospective nature of the work many years later.

Importance

The submission is an important contribution as it highlights important etiological trends in PDrP and advocates for further research towards preventing streptococcal and polymicrobial infections.

Abstract

Appropriate wording and description.

Throughout the manuscript, period has been replaced by a comma probably related to a software issue – example please change 1,6% to 1.6%

Please reword “We disclosed significant trends to a decrease” and “The last three decades have contemplated a growing involvement”.

Throughout the manuscript, please replace “associated a” with “was associated with a”

Introduction

Written well. No significant changes recommended.

Please reword the last paragraph as it is wordy and unclear “Our main interest was to pursue a clinical perception that advances in the prevention and management of PDrP could have modified the clinical scenery of PD-related peritonitis, conferring streptococci a progressively dominant role, with potential consequences for the clinical presentation, management and outcomes of these infections.”

Methods

The general design, study variables and outcomes are appropriate.

Please delete “(primary objectives too)”

Is information regarding methicillin resistance available for Staphylococcus aureus? If so, please include it. If not available, please note that it is not available.

Describe outpatient diagnosis, and management of PDrP and if any changes have been made over the last 30 years. Since hospital admission for PDrP is indicated for high-risk patients, infections with an aggressive clinical presentation, treatment failure after oriented antimicrobial therapy and isolation of microorganisms demanding in centre management – what is the management for those that don’t meet these criteria? Are cultures obtained prior to empiric therapy as an outpatient? This may have a significant impact on epidemiology, etiology, presentation and outcomes.

Are chlorhexidine baths/ wipes used for any patients?

Please cite SPSS per guideline

Results

Four hundred and sixty-two patients (52.6%) suffered at least one episode of PDrP. This appears to be inconsistent.

To clarify, did 462 out of 878 patients evaluated have a total of 1061 episodes of PDrP? If so, could this be stated more clearly for the reader?

Antibiotic association should be delineated better. Is the implication deviation from standard practice?

Clinical outcomes are described in an excellent manner.

Discussion

Well described. No significant changes recommended.

Given the suspicion for enteric source due to Streptococcal, polymicrobial and Gram negative infections even after exclusion of an obvious surgical etiology, it may help to include how many of these instances had negative abdominal imaging (if information available).

With streptococcal PDrP, could presentation be delayed due to a sub-acute onset of illness and therefore present with a higher PD fluid cell count?

Acknowledge limitation that therapy used may be broader than necessary for streptococcal PDrP. Since this paper evaluates outcomes, this may be important.

Limitations:

Overall, well described.

References:

Appropriate.

Table 1-7; Figure 1

Clear and appropriately defined. For figure, can consider trend line chart.

Reviewer #2: The authors have studied the incidence and characteristics of peritonitis over a 30-year period.

The paper is interesting, but there are a few problems with the analysis.

1) The authors seem to fall between two stools in their methodology. The stated aim is particular focus on streptococcal infections, but they present data on all common forms of peritonitis, so it is only in the discussion that any real differences in approach appear. Consider altering the focus to a general overview of the peritonitis experience (the most interesting choice for this reviewer). I am not clear why they have chosen streptococci as the primary subject of investigation. If they wish to maintain the focus on streptococci, perhaps just comparing streptococci with “all others”.

2) The incidence of streptococcal infections is more or less unchanged, varying between 0,063 and 0,122/year for a relatively small number of cases. This is confirmed by the joinpoint analysis. The title is thus somewhat misleading, and could really be called “The falling incidence of staphylococci…..”. The relative incidence of streptococci and staphylococci is uninteresting in this respect.

3) Several of the variables do not seem to be normally distributed, e.g. Charlson Comorbidity score, peritoneal cell count, time to remission or catheter removal, but are presented and analysed as such in the multivariate analyses.

4) Some of the statistics “feel” wrong. E.g an unchanged Strept. peritonitis incidence translates into a 77% increase in Table 4. A more or less identical time to remission is significant in Table 6. Please check for common statistical errors (maybe using professional statistical assistance), e.g. failing to correct for multiple comparisons, using parametric analyses for non-parametric variables, treating categorical variables as continuous variables.

Minor Comments

1) Page 8 ” including 25 patients (15,8%) with more than two monobacterial streptococcal PDrP during follow-up (range 3-7).” Slightly unclear. 3-7 months? What about patients with two monobacterial streptococcal PDrP during follow-up?

2) Page 9. “Median delays to the first episode of PDrP were 8 (interquartile range 3 to 17) (overall) and 12 months (interquartile range 5 to 24)(streptococcal PDrP).” Does “overall” include or exclude streptococcal PDrP?

3) Page 9 “On the other hand, at least one streptococcal strain was isolated in 78 polymicrobial infections”. Table 2 seems to state 116 patients. Unclear, please clarify.

4) “No patient denied consent”. In this long-term retrospective study, requirement for patient consent was presumably waived?

5) Table 4. Four outcomes are defined in the Methods, but only three are presented. Were the outcomes censored for each other?

6. PLOS authors have the option to publish the peer review history of their article (what does this mean?). If published, this will include your full peer review and any attached files.

Reviewer #1: No

Reviewer #2: No

---

## [Author Response · Author response to Decision Letter 0]

21 Nov 2020

1) Editorial Board

- We have adapted the format of the manuscript to the requirements of the Journal. Figure 1 has been changed to a TIF format, as requested.

- We now state in the paper that oral informed consent was obtained only for the minority of patients who were accessible at the time of the study. In these cases, consent was approved by the Ethical Committee, and registered in the clinical records of the patients. For nonavailable patients, permission was given by the ethical Committee, under the condition of full de-identification of the cases.

- Regarding the question of open access to the data sets, sorry, this is not possible under Spanish or Galician regional data protection acts, even if the data bases are de-identified. However, this information can be obtained on request, needing only a previous approval by the local ethical committee. This information should be requested through the Galician Health System (Servicio Galego de Saúde, Sergas), in this address:

Dr. Celia Cal Purriños

Secretary of the Research Ethics Committee of the A Coruña-Ferrol Area

Rede Galega de Comités

Xerencia do Servizo Galego de Saúde

Edificio Administrativo San Lázaro s/n

15703 SANTIAGO DE COMPOSTELA (Spain)

Mail: ceic@sergas.es

Please, remember that the code number of the study for the Ethical Committee is 2020/190.

2) Reviewer #1

We warmly thank the kind general comments of the reviewer. Regarding the concerns and suggestions:

- Overview

The authors have conducted a 30-year retrospective chart review to evaluate trends in incidence of PD-related peritonitis. They have found decreasing trends in staphylococci and Gram negative bacterial infections and an increase in streptococcal and polymicrobial infection. There wasn’t a difference in predictive risk factors for Streptococcal PDrP but had more favorable outcomes. There were no differences amongst species of Streptococci. This is information that is useful to clinicians, infection prevention and improving quality of care. The authors have described the various aspects of this study succinctly and performed an excellent review of why this is an issue of practical importance. They also highlight, the potential difficulties with interpreting this data due to the retrospective nature of the work many years later.

Importance

The submission is an important contribution as it highlights important etiological trends in PDrP and advocates for further research towards preventing streptococcal and polymicrobial infections.

Abstract

Appropriate wording and description.

Throughout the manuscript, period has been replaced by a comma probably related to a software issue – example please change 1,6% to 1.6%

We have replaced commas by periods to express decimals, as requested

- Please reword “We disclosed significant trends to a decrease” and “The last three decades have contemplated a growing involvement”.

Throughout the manuscript, please replace “associated a” with “was associated with a”

Thank you for the style corrections in the Abstract and text. We have implemented them, as suggested

- Introduction

Written well. No significant changes recommended.

Please reword the last paragraph as it is wordy and unclear “Our main interest was to pursue a clinical perception that advances in the prevention and management of PDrP could have modified the clinical scenery of PD-related peritonitis, conferring streptococci a progressively dominant role, with potential consequences for the clinical presentation, management and outcomes of these infections.”

We have reworded the last paragraph of the Introduction, as requested. We fully agree that the original one was wordy and confusing

- Methods

The general design, study variables and outcomes are appropriate.

Please delete “(primary objectives too)”

In Method, we have deleted “primary objectives too”, as requested

- Is information regarding methicillin resistance available for Staphylococcus aureus? If so, please include it. If not available, please note that it is not available.

In Method-Study Variables and Strategies, second paragraph, we have included a sentence indicating that 4 episodes of Staph. aureus infection were caused by methicillin-resistant strains. This small group was not a subject of individualized analysis, as now stated.

- Describe outpatient diagnosis, and management of PDrP and if any changes have been made over the last 30 years. Since hospital admission for PDrP is indicated for high-risk patients, infections with an aggressive clinical presentation, treatment failure after oriented antimicrobial therapy and isolation of microorganisms demanding in centre management – what is the management for those that don’t meet these criteria? Are cultures obtained prior to empiric therapy as an outpatient? This may have a significant impact on epidemiology, etiology, presentation and outcomes.

In Method-Study Variables and Strategies, last two paragraphs, we have made changes to explain better the diagnostic procedures and outpatient management criteria, as requested

- Are chlorhexidine baths/ wipes used for any patients?

Sorry, our procedures do not include clorhexidine baths/wipes

- Please cite SPSS per guideline

We now cite the software as IBM-SPSS 

- Results

Four hundred and sixty-two patients (52.6%) suffered at least one episode of PDrP. This appears to be inconsistent.

To clarify, did 462 out of 878 patients evaluated have a total of 1061 episodes of PDrP? If so, could this be stated more clearly for the reader?

We have changed the text at the start of Results, to clarify the percentages, as suggested

- Antibiotic association should be delineated better. Is the implication deviation from standard practice?

Regarding antibiotic associations, the main ones are depicted as a footnote in Table 5. We were not more specific because there were many different combinations. These associations were within common standards, but we used different betalactamics, amynoglycosides, glycopeptides and so on, and there were changes in the course of treatment in some cases. In our opinion, describing all this in detail does not have a major interest, and could result in a really wordy paragraph. We have decided not to change this, but will reconsider, if you request us to do so.

- Discussion

Well described. No significant changes recommended.

Given the suspicion for enteric source due to Streptococcal, polymicrobial and Gram negative infections even after exclusion of an obvious surgical etiology, it may help to include how many of these instances had negative abdominal imaging (if information available).

Sorry, information on how many patients had a negative (normal) abdominal imaging is not available, although we could say “a majority”. This is an interesting question. We routinely perform abdominal imaging studies in all cases of polymicrobial peritoneal infection with involvement of anaerobics or Enterobacteriaceae, as also in monobacterial infections by Enterobacteriaceae or Staph. aureus following a torpid or aggressive clinical course. In our experience, both ultrasonography and CT scan offer a relatively low sensitivity to detect surgical conditions. However, we continue to perform these studies, because they are very orientative, when positive.

- With streptococcal PDrP, could presentation be delayed due to a sub-acute onset of illness and therefore present with a higher PD fluid cell count?

Regarding clinical presentation of streptococcal PDrP, our experience, clearly reflected in the manuscript, suggests that these infections have an explosive, rather than subacute onset. The fact that they appear (not so much) later than other infections may reflect the earlier appearance of staphylococcal infections, which are often a consequence of a poor technique performance.

- Acknowledge limitation that therapy used may be broader than necessary for streptococcal PDrP. Since this paper evaluates outcomes, this may be important.

We have included a new sentence in the Limitations paragraph of the Discussion, to recognize that therapy used for streptococcal infections may have been broader than necessary, as requested.

- Table 1-7; Figure 1

Clear and appropriately defined. For figure, can consider trend line chart.

As refers to Figure 1, we have tested the lines and the bars options, and the latter looked clearly better

Reviewer #2

 We also thank the general comments of this reviewer. Regarding the specific comments:

- The authors seem to fall between two stools in their methodology. The stated aim is particular focus on streptococcal infections, but they present data on all common forms of peritonitis, so it is only in the discussion that any real differences in approach appear. Consider altering the focus to a general overview of the peritonitis experience (the most interesting choice for this reviewer). I am not clear why they have chosen streptococci as the primary subject of investigation. If they wish to maintain the focus on streptococci, perhaps just comparing streptococci with “all others”.

The reason why we had a particular focus on streptococcal infections was a perception, supported by preliminary data, that the relevance of these microorganisms as causative agents of PDrP had increased progressively during the last decades. However, we agree with the reviewer that this was not just a question of increasing streptococcal infections (in fact, this was close, but did not reach statistical significance), but also of a reciprocal decline in the incidence of staphylococcal and Gram negative infections. This is why a general analysis was necessary, to present adequately the question. We have modified the last paragraph of the Introduction, to be more inclusive.

- The incidence of streptococcal infections is more or less unchanged, varying between 0,063 and 0,122/year for a relatively small number of cases. This is confirmed by the joinpoint analysis. The title is thus somewhat misleading, and could really be called “The falling incidence of staphylococci…..”. The relative incidence of streptococci and staphylococci is uninteresting in this respect.

The incidence of streptococcal infections showed a clear trend (close to significance by joinpoint regression) to an increase. You should also consider that polymicrobial infections (with isolation of streptococci in 58.6% of the cases) also increased during the study period. The remarkable point is that, while staphylococcal and Gram negative infections decreased (both in absolute and relative terms), streptococcal infections (and their polymicrobial “associates”) tended to increase, significantly (relative) or close to significance (absolute). This facts demand explanation and, likely, changes in procedures.

- Several of the variables do not seem to be normally distributed, e.g. Charlson Comorbidity score, peritoneal cell count, time to remission or catheter removal, but are presented and analysed as such in the multivariate analyses.

Regarding the distribution of some variables, this question was decided by our epidemiologist (SPD). It is true that some variables (e.g. Charlson, GFR or peritoneal cell count) did exhibit some deviation from normal distribution. However, the deviation was not marked, and the sample was large enough to permit analyses by parametric tests. The only exception was C reactive protein, which exhibited a markedly abnormal distribution (Table 1), and had to be categorized for analysis.

- Some of the statistics “feel” wrong. E.g an unchanged Strept. peritonitis incidence translates into a 77% increase in Table 4. A more or less identical time to remission is significant in Table 6. Please check for common statistical errors (maybe using professional statistical assistance), e.g. failing to correct for multiple comparisons, using parametric analyses for non-parametric variables, treating categorical variables as continuous variables.

We have checked the potential errors in statistic analysis, and believe that the presented figures are correct. In the case of Streptococcus incidence, the change was not significant, but close to, in univariate analysis. Table 4 presents adjusted results (multivariate), and comparisons are made against the period 1990-99 (lowest incidence), while joinpoint regression analyzes the whole period. On the other side, significance in line 1 of Table 6 (time to remission) represents the result of the omnibus ANOVA test for all the categories (by the way, p value was 0.033 and not 0.023, sorry). Please, note that there is no figure in bold, indicating that, in post hoc analyses, streptococci were not the source of this significant difference.

- Page 8 ” including 25 patients (15,8%) with more than two monobacterial streptococcal PDrP during follow-up (range 3-7).” Slightly unclear. 3-7 months? What about patients with two monobacterial streptococcal PDrP during follow-up?

In page 8, we have clarified the figures for repeat streptococcal infections, according to the request of the reviewer. We hope this is adequately clarified

- Page 9. “Median delays to the first episode of PDrP were 8 (interquartile range 3 to 17) (overall) and 12 months (interquartile range 5 to 24)(streptococcal PDrP).” Does “overall” include or exclude streptococcal PDrP?

In the last sentence of the previous paragraph (median delays to the first episode of PDrP), overall includes streptococcal infections

- Page 9 “On the other hand, at least one streptococcal strain was isolated in 78 polymicrobial infections”. Table 2 seems to state 116 patients. Unclear, please clarify.

In page 9, the disagreement is ony apparent: 116 different strains of streptococci were isolated during 78 episodes of polymicrobial peritonitis. It was not uncommon to isolate 2 or 3 (even 4, in one case) different streptococcal strains in the same episode

- “No patient denied consent”. In this long-term retrospective study, requirement for patient consent was presumably waived?

The reviewer is fully right about the question of informed consent. Only a minority of patients were accessible at the time of the study. For the others, we obtained a permission of the Ethical Committee. We have made a change in the last paragraph of Method-General Design, to clarify this.

- Table 4. Four outcomes are defined in the Methods, but only three are presented. Were the outcomes censored for each other?

The four outcomes mentioned are depicted at the bottom of Table 6

---

## [Decision Letter · Decision Letter 1]

2 Dec 2020

PONE-D-20-22429R1

The Growing Role of Streptococci as Causative Agents of Peritoneal Dialysis-related Peritonitis. A Longitudinal Analysis

PLOS ONE

Dear Dr. Fontan,

Thank you for submitting your manuscript to PLOS ONE. After careful consideration, we feel that it has merit but does not fully meet PLOS ONE’s publication criteria as it currently stands. Therefore, we invite you to submit a revised version of the manuscript that addresses the points raised during the review process.  Specifically, one reviewer is at odds with your conclusions.  I would like you to review these comments and indicate whether or not the minor changes proposed are reasonable. 

Please submit your revised manuscript within two weeks. If you will need more time than this to complete your revisions, please reply to this message or contact the journal office at plosone@plos.org. Please include the following items when submitting your revised manuscript:

We look forward to receiving your revised manuscript.

Kind regards,

Sean Reid

Academic Editor

PLOS ONE

Reviewers' comments:

Reviewer's Responses to Questions

**Comments to the Author**

1. If the authors have adequately addressed your comments raised in a previous round of review and you feel that this manuscript is now acceptable for publication, you may indicate that here to bypass the “Comments to the Author” section, enter your conflict of interest statement in the “Confidential to Editor” section, and submit your "Accept" recommendation.

Reviewer #1: All comments have been addressed

Reviewer #2: All comments have been addressed

2. Is the manuscript technically sound, and do the data support the conclusions?

Reviewer #1: Yes

Reviewer #2: Yes

3. Has the statistical analysis been performed appropriately and rigorously? 

Reviewer #1: Yes

Reviewer #2: Yes

4. Have the authors made all data underlying the findings in their manuscript fully available?

Reviewer #1: No

Reviewer #2: Yes

5. Is the manuscript presented in an intelligible fashion and written in standard English?

Reviewer #1: Yes

Reviewer #2: Yes

6. Review Comments to the Author

Reviewer #1: (No Response)

Reviewer #2: The authors have responded satisfactorily to previous comments.

I have only one remaining issue.

The bottom line is that no significant change in incidence of streptococcus peritonitis has ocurred. Thus the title "The Growing Role of Streptococci as Causative Agents of Peritoneal Dialysis-related Peritonitis." and the conclusion "Streptococci have acquired an increasing role as causative agents" are misleading.

It is not usual to present an insignificant trend as a conclusion Indeed, in the opinion of this reviewer, a more scientific title would be “Unchanged incidence of streptococcal peritonitis over a 30-year period”, but at the very least it should be emphasised that it is relative and not absolute incidence that is being discussed. As previously stated, the emphasis on relative incidence is not very interesting.

7. PLOS authors have the option to publish the peer review history of their article (what does this mean?). If published, this will include your full peer review and any attached files.

Reviewer #1: No

Reviewer #2: No

---

## [Author Response · Author response to Decision Letter 1]

3 Dec 2020

Thank you for considering the revised version of our study on “The Growing Role of Streptococci as Causative Agents of Peritoneal Dialysis-related Peritoneal Infections. A Longitudinal Analysis” (PONE-D-20-22429). We are now submitting a secondly revised version of the manuscript, addressed to answer the remaining concerns. According to the instructions, we include a marked-up copy of the paper with the modifications introduced (red for deletions, blue for additions), as also a clean copy. We shall now state in detail the modifications introduced and the answers to the reviewers.

1) Editorial Board

We understand that there is no new specific concern or query.

2) Reviewer #1

As in the previous case, we understand that there is no specific concern or query. We thank the reviewer for his/her contributions.

3) Reviewer #2

 We also thank the relevant concerns manifested by this reviewer. First, we want to state that, to our belief, the combination of clear trends to an increase in the incidence of streptococcal monobacterial (even if not significant!), and polymicrobial infections (significant and with a prominent involvement of streptococci) permits to speculate that streptococcal infections may have partly occupied the niche left by staphylococci and GNB. In any case, we also feel that the changes proposed by the reviewer are very reasonable. These changes downplay, but do not distort the original conclusions of the paper, and we have decided to follow the recommendation. Consequently, we have modified the title of the manuscript, the Conclusions of the Abstract, the Discussion (just one word deleted to downplay the contention) and the Summary, at the end of the Discussion. We hope that these new changes to the presentation are satisfactory.

Again, we thank the reviewers for their attention, their kind comments and, most importantly, for their interesting and consequential concerns and recommendations. We hope that the secondly revised manuscript is now satisfactory, but will readily consider any new suggestion or recommendation.

Sincerely

Miguel Pérez Fontán

---

## [Editor Report · Decision Letter 2]

8 Dec 2020

Long-Term Trends in the Incidence of Peritoneal Dialysis-Related Peritonitis Disclose an Increasing Relevance of Streptococcal Infections. A longitudinal Study

PONE-D-20-22429R2

Dear Dr. Fontan,

We’re pleased to inform you that your manuscript has been judged scientifically suitable for publication and will be formally accepted for publication once it meets all outstanding technical requirements.

Kind regards,

Sean Reid

Academic Editor

PLOS ONE
---

## [Editor Report · Acceptance letter]

10 Dec 2020

PONE-D-20-22429R2 

Long-Term Trends in the Incidence of Peritoneal Dialysis-Related Peritonitis Disclose an Increasing Relevance of Streptococcal Infections. A longitudinal Study 

Dear Dr. Pérez Fontán:

I'm pleased to inform you that your manuscript has been deemed suitable for publication in PLOS ONE. Congratulations! Your manuscript is now with our production department. 

Kind regards, 

on behalf of

Dr. Sean Reid 

Academic Editor

PLOS ONE